# Citation Accuracy: A Case Study on Definition of the Grey Water Footprint

**Libor Ansorge *** and **Lada Stejskalová**

T. G. Masaryk Water Research Institute, Podbabská 2582/30, 16000 Prague, Czech Republic
* Correspondence: libor.ansorge@vuv.cz; Tel.: +420-220-197-285

**Abstract:** Citing sources is an important part of any research paper. A number of studies in the past have dealt with discrepancies or errors in citations. The citation errors range from several percent to tens of percent. Although in most cases, these are minor formal inconsistencies that do not prevent tracing the source used, there are cases where the citations are incorrect or refer to non-existent articles. In this study, an analysis of the citation of the grey water footprint definition was conducted. The water footprint concept was introduced in 2002 as a methodology for the description of quantity aspects linked to water use in the life cycle. The grey water footprint, which represents the quality aspects of water use, was added to the water footprint concept later on. In this study, almost 300 articles that provide a definition of the grey water footprint and are indexed in the Scopus database were reviewed. More than two-thirds of the articles added the definition by citing the source. Only 3.5% of the citing articles contained an incorrect citation that could be considered a significant error. Although this is a low number, these significant errors have been appearing only in recent years. This suggests the possibility that the percentage of errors could gradually increase as the use of grey water footprint expands in practice. In the first period (up to 2017), only the grey water footprint originators are cited. From 2017 onwards, papers not written by the originators of the grey water footprint idea are also cited.

**Keywords:** citation accuracy; citation analysis; grey water footprint

## 1. Introduction

References are an important part of the scientific literature. They identify sources of primary literature and help authors refer to the originators of the ideas and texts from which their own work draws. Citations of source documents also help the reader place the article in the context of current knowledge. In general, there is a number of reasons for citing used sources. Correct citation of all used sources is also the main condition of ethical writing. The main purpose of citations is to enable the tracing of original scientific sources. Inaccurate or erroneous citations can be displeasing for the original author, misleading for the reader, and, if continually repeated, may turn inaccuracies into conventional knowledge [1]. However, even researchers are only human and, therefore, make mistakes, including mistakes in referencing the sources used. Sauvayre [2] showed that both junior and senior researchers make such mistakes. There are many types of referencing errors. Usually, we distinguish (i) errors in reference writing (citation errors), which cause difficulties in tracing the source reference, and (ii) errors in the quotation content (quotation errors) that change the meaning of the quoted text. Inshakova and Pankeev [3] state that the most frequent errors are related to quotation content. Lazonder and Janssen [4] "postulate that the occurrence and seriousness of quotation errors depend on how detailed or specific the content of a referenced source is described".

Many authors have dealt with the analysis of citation errors. These analyses were focused on citation errors either of a specific work [5] or journals [6–8]. A number of works deal with citation errors in specific areas of research, especially in healthcare [4,9–13], errors in specific types of documents [14], or in specifically focused studies [15,16].

*1.1. Grey Water Footprint History*

Grey water footprint (GWF) is a part of the water footprint concept, which was introduced in 2002 [17]. However, at that time the water footprint contained only the blue and green parts. The grey water footprint was added to the concept several years later. According to Hoekstra [18] (the originator of the water footprint concept), the green, blue, and grey water footprints were presented in one coherent framework for the first time in a book in 2008 [19]. It is an interesting statement because in the article by Hoekstra and Chapagain from 2007 [20] are texts " ... *water footprint has three components: the blue, green and grey water footprint*", and " ... *the grey water footprint component refers to the volume of water required to dilute pollutants to such an extent that concentrations are reduced to the agreed maximum acceptable levels* ... ". The grey water footprint principle was introduced a few years before that, in the study "The Water Footprint of Cotton Consumption" in 2005 [21] and the follow-up scientific article "The Water Footprint of Cotton Consumption: An Assessment of the Impact of Worldwide Consumption of Cotton Products on the Water Resources in the Cotton Producing Countries" in 2006 [22].

The methodological issues of the grey water footprint were then developed by the international working group, whose work resulted in a number of refinements, including consideration of the quality of the water abstracted, and a multi-level approach to distinguish different levels of detail in the assessment of the grey water footprint of diffuse pollution. The work of this group has been reflected in the Water Footprint Assessment Manual [23] issued in 2011, which is the basic water footprint methodology document. There, the grey water footprint is defined as "*the volume of water that is required to assimilate waste, quantified as the volume of water needed to dilute pollutants to such an extent that the quality of the ambient water remains above agreed water quality standards*".

In the years 2012–2013, an international expert group prepared the grey water footprint guidelines for the application so-called Tier 1, providing additional practical help in assessing the grey water footprint for a variety of chemicals [24]. This guideline defines the grey water footprint in a slightly different way: "*The grey water footprint is defined as the volume of freshwater that is required to assimilate a load of pollutants to a freshwater body, based on natural background concentrations and existing ambient water quality standards. The GWF is calculated as the volume of water that is required to dilute pollutants (chemical substances) to such an extent that the quality of water remains above agreed ambient water quality standards*".

In the context of correct citation of originators of grey water footprint, we can conclude that these documents could be cited:

- The study of "The water footprint of cotton consumption" [21], which first used the principles of grey water footprint;
- The scientific article "The water footprint of cotton consumption: An assessment of the impact of worldwide consumption of cotton products on the water resources in the cotton producing countries" [22], which introduced the principles of grey water footprint to the scientific community;
- The scientific article "The water footprints of Morocco and the Netherlands: Global water use as a result of domestic consumption of agricultural commodities" [20], which first introduced the term "grey water footprint";
- The book "Globalization of Water: Sharing the planet's freshwater resources" [19], which is marked as the first coherent framework that introduced the grey water footprint;
- The "Water footprint assessment manual" [23], which is the main methodological water footprint document;
- The guidelines "The grey water footprint accounting: Tier 1 supporting guidelines" [24], which improve the definition of the grey water footprint.

*1.2. The Aim of the Study*

The main impetus for writing this article came in September 2022, when one of the authors came across with two articles, within one week, that incorrectly cited the grey water footprint definition. An incorrect definition of the grey water footprint can lead to

the derivation of incorrect equations for its calculation, which has been documented in a number of articles that have been mapped by the first author in the past [25]. Ultimately, this can lead to an incorrect assessment of the impacts of water use as assessed by the water footprint methodology.

In this study, we focused on citation errors of a specific idea. The aim of the study was to analyze errors in citing the grey water footprint definition. Specifically, answers to the following research questions were sought:

1. What is the percentage of citations of wrong source articles in the current scientific literature?
2. What percentage of citations do not refer to the originators of the idea, but to another source?

The first research question is interesting in terms of citation accuracy. The second question can show whether the grey water footprint is already being incorporated into the standard indicators and citing authors no longer look for the originator of the idea, but only cite the source from which they draw their infomation. R. K. Merton termed this process as "obliteration by incorporation" [26], whereby the original ideas and their originators are incorporated into common knowledge and their followers are cited in more recent works.

## 2. Materials and Methods

In this paper, we use terms defined by Smith and Cumberledge [27] as follows:

- '*article*' refers to the primary article in which the citation occurs;
- '*reference*' refers to the book, article, etc. that is being cited;
- '*citation*' refers to an individual instance of citing a reference.

Data collection and analysis followed several steps, which are shown in Figure 1 below and further described in the following paragraphs.

### 2.1. Materials Preparation

The study included articles contained in the Scopus database [28] that have the term "Grey water footprint" in the title, keywords, or abstract, and have English listed as the language. Data collection was conducted on 23 October 2022, and subsequently updated on 26 November 2022, using the search query:

TITLE-ABS-KEY ("grey water footprint" OR "gray water footprint") AND LANGUAGE ("English")

A total of 359 records were found, which were exported in csv format and saved in Google Table. The following data were exported: authors, title, year, source title, digital object identifier (DOI), link, and Scopus identifier (EID). For each article found, an analysis was made as to whether it was written by one of the authors of the water footprint concept or the grey water footprint idea, i.e., A. Y. Hoekstra, M. M. Mekonnen, A. K. Chapagain, M. M. Aldaya, H. H. G. Savenije, or R. Gautam.

For all records found, an attempt was made to obtain the full texts of these articles either on the publisher's website, by searching the Internet (e.g., ResearchGate), or through electronic communication with authors. For 9 records, it was not possible to obtain the full text of the article so these records were omitted from further processing.

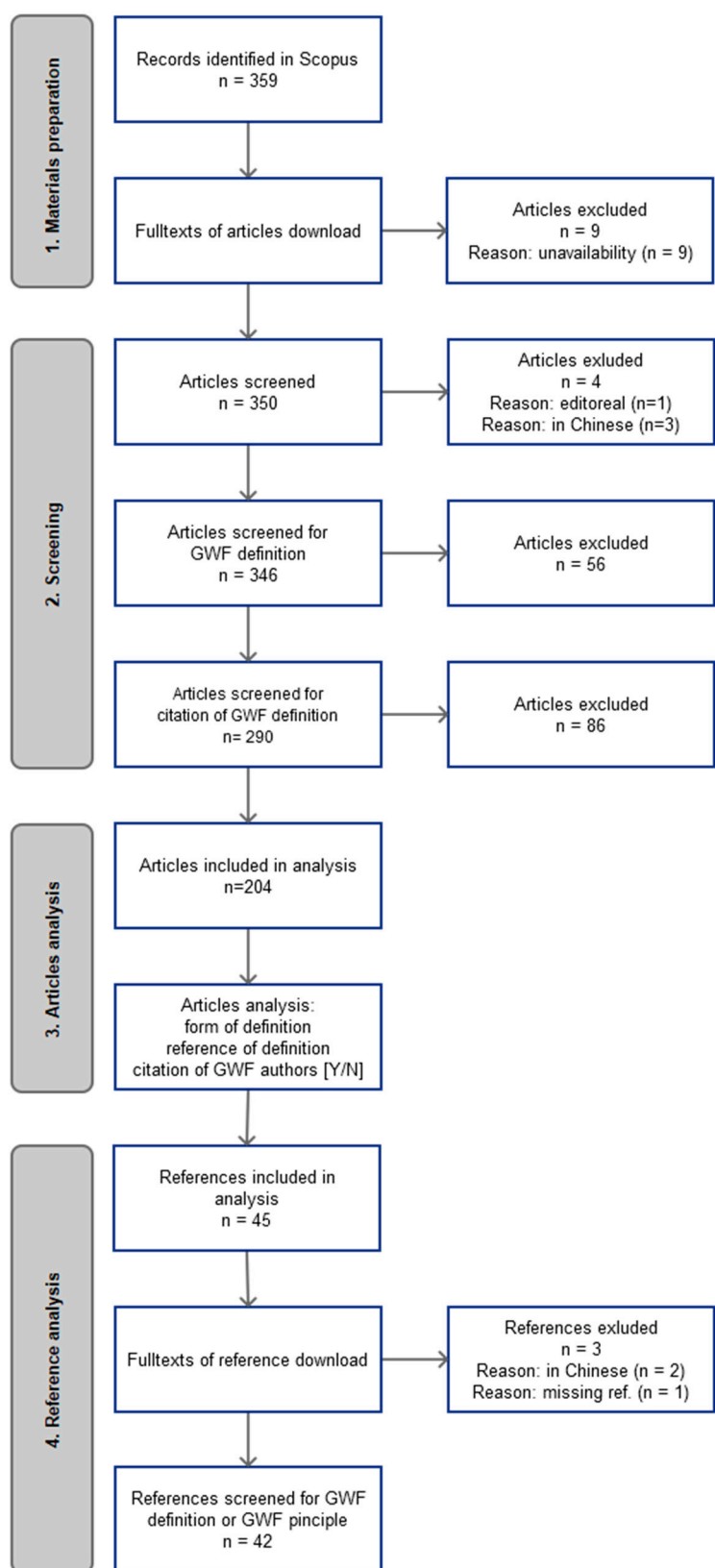

**Figure 1.** Methodological scheme for analysis.

### 2.2. Screening

In the next step, articles were screened to see if they were scientific articles dealing with the water footprint. A total of 350 articles were included in the screening, of which

4 articles were discarded. One article was excluded because it was an editorial for the scientific conference proceeding; the other three articles were excluded due to being written in Chinese and available only in PDF.

In the next screening step, 56 articles that did not contain a verbal definition of water footprint were excluded. That means that each article was searched for text identical or similar to the definitions described in Section 1.1. Articles that did not contain the textual description corresponding to the textual definition according to the Water Footprint Assessment Manual [23] or the description corresponding to the principle of the grey water footprint described in the article by Chapagain et al. [22] were excluded from further processing. In doing so, it was not important how accurate the definition was, but whether the text used matched the word definition. The formula for calculating the grey water footprint was not considered as a definition. Therefore, text such as "*the GWF refers to amount of water* . . . " or "*the GWF is defined as the volume of water* . . . " or "*the GWF representing the volume of water* . . . " and similar were considered as verbal definitions of the grey water footprint. Conversely, text verbally describing the formula for calculating the grey water footprint was not considered a verbal definition.

In the last step of the screening, another 86 articles were excluded. These articles did mention the definition of the grey water footprint but did not cite the source of the definition. Thus, 204 articles were included in the analysis that provided a verbal definition of the grey water footprint and cited the source of the definition.

*2.3. Article Analysis*

For every analyzed article containing a citation of the water footprint definition, the definition used in the article, the reference, and the DOI or URL of the cited (source) document were copied into a spreadsheet table. If more than one reference document was cited, then the oldest reference in which the grey water footprint definition was found was used. A total of 49 different source documents were identified and cited in relation to the grey water footprint definition.

*2.4. Reference Analysis*

In the first step, the full texts of the source documents were obtained. Three records were excluded from further processing: one article was excluded due to a missing reference in the references, and two articles were excluded due to a citation of a Chinese source in which the existence of a grey water footprint definition could not be verified.

In the second step, the source documents were analyzed to see if they contained the definition of the grey water footprint in accordance with the Water Footprint Assessment Manual [23], with the guidelines for Tier 1 [24] or the principle of the grey water footprint described in the article by Chapagain et al. [22].

**3. Results**

*3.1. Citation Errors*

In this study, 204 citations of the grey water footprint definition were checked (see Table 1). For two records, it was not possible to analyze the existence of the definition in the reference for linguistic reasons, as they were in Chinese. One record could not be analyzed at all because the article used a numeric citation style and the reference information was completely missing in the references. This is considered a serious error because the reader cannot locate the source document. Another six cases, where the grey water footprint definition or principle is not stated in the cited document, can be considered as containing a serious error. Serious errors, thus, represent 3.5% of citations.

**Table 1.** Results of analysis of the GWF definition/GWF principle citation.

| Description →<br>Year ↓ | Missing Definition in Reference | Missing Reference in Article | Cited Source in Chinese | GWF Definition in Reference | GWF Principle in Reference | Total |
|---|---|---|---|---|---|---|
| 2023 | | | | 2 | | 2 |
| 2022 | 2 | | 1 | 34 | 2 | 39 |
| 2021 | 2 | | | 32 | | 34 |
| 2020 | 1 | 1 | 1 | 25 | 2 | 30 |
| 2019 | | | | 22 | 2 | 24 |
| 2018 | | | | 19 | 1 | 20 |
| 2017 | | | | 16 | 3 | 19 |
| 2016 | | | | 13 | | 13 |
| 2015 | | | | 5 | 1 | 6 |
| 2014 | 1 | | | 8 | 1 | 10 |
| 2013 | | | | 3 | | 3 |
| 2012 | | | | 2 | | 2 |
| 2011 | | | | 2 | | 2 |
| 2010 | | | | | | |
| Total | 6 | 1 | 2 | 183 | 12 | 204 |

The citation of a book review in which a review of books dealing with the water footprint is carried out can be considered as a less serious error, because the reader will reach the correct source document through this book review. Book reviews have been cited three times—twice the book review of the Water Footprint Assessment Manual [23] and once the book review of Hoekstra and Chapagain [19].

*3.2. Who Is Cited Analysis*

Table 2 (Figure 2) shows the number of references to articles written by the authors of the water footprint concept and the grey water footprint idea, i.e., A. Y. Hoekstra, M. M. Mekonnen, A. K. Chapagain, M. M. Aldaya, H. H. G. Savenije, or R. Gautam, with comparison to the number of references to articles written by other authors. A total of 9.85% of references are so-called "empty" references that do not contain an original idea under investigation, but strictly refer to other studies to substantiate their claim [29].

**Table 2.** Who is cited during the years.

| Year | Cited Reference Authored by Originators of GWF | Cited Other Reference | Total | % of Cited Reference by Originators of GWF on Total |
|---|---|---|---|---|
| 2023 | 2 | | 2 | 100% |
| 2022 | 28 | 11 | 39 | 71.79% |
| 2021 | 32 | 2 | 34 | 94.12% |
| 2020 | 26 | 3 | 29 | 89.66% |
| 2019 | 22 | 2 | 24 | 91.67% |
| 2018 | 19 | 1 | 20 | 95% |
| 2017 | 18 | 1 | 19 | 94.74% |
| 2016 | 13 | | 13 | 100% |
| 2015 | 6 | | 6 | 100% |
| 2014 | 10 | | 10 | 100% |
| 2013 | 3 | | 3 | 100% |
| 2012 | 2 | | 2 | 100% |
| 2011 | 2 | | 2 | 100% |
| Total | 183 | 20 | 203 | 90.15% |

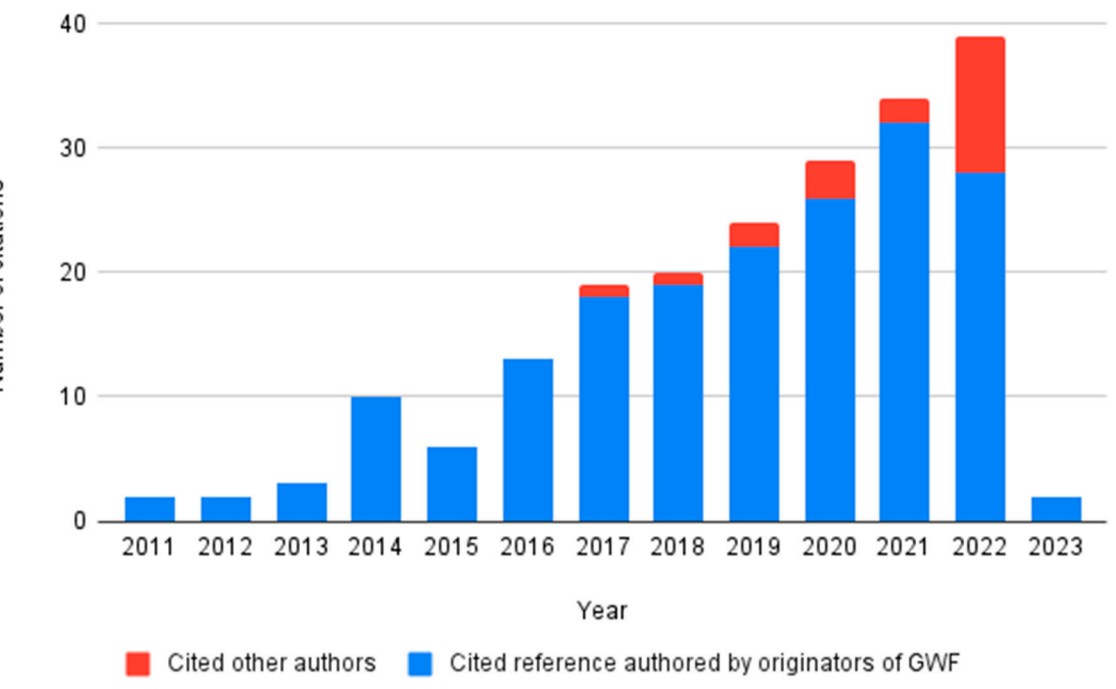

**Figure 2.** Cited reference authored by the originators of the GWF and by other authors.

Table S1 in supplementary materials shows references being cited in the analyzed articles. The Water Footprint Assessment Manual [23], which is the basic methodological document of the water footprint and also defines the grey water footprint, was cited the most often (a total of 106 times). Out of these 106 cases of citation, it was cited five times together with the article by Chapagain et al. [22], where the grey water footprint principle was first introduced. However, in two cases, the Water Footprint Assessment Manual [23] was cited through a book review citation. The article by Chapagain et al. [22], in addition to five joint citations with the Water Footprint Assessment Manual, was cited in six other cases and, thus, was the second most cited reference. The third most cited reference with nine citations is the guidelines for the application of the Tier 1 approach by Franke et al. [24]. The book Globalization of Water: Sharing the Planet's Freshwater Resources [19] ranked (with seven citations) as the fourth–fifth most cited reference. This book is referred to as the first to present the grey water footprint as part of the water footprint concept by A. Hoekstra himself [18], and also by other authors e.g., [30].

## 4. Discussion

Among the 86 articles excluded from the analysis in the last step of the screening, there were a number of cases where the Water Footprint Assessment Manual [23] was cited in one sentence and the definition of the individual components of the water footprint was given in a subsequent separate sentence. Nevertheless, we decided to exclude these cases from the analysis, in order to maintain maximum objectivity. If we also decided to include in the definition citation cases where the citation is given in another sentence with a logical link to the definition itself, then it would be necessary to decide on the "permitted distance" of the citation, i.e., whether to include the citation only from the directly adjacent sentence, or the citation within the given paragraph as well, etc. Deciding whether to count the quotation as correct or incorrect would then be very subjective.

For the same reason, i.e., limiting subjective opinions, we decided not to assess the "accuracy" of the adopted definition, even though in many cases it is clearly an erroneous adoption. A typical error, which was repeated in several articles, consisted in confusing "waste load" with "wastewater" [31–33] or "sewage" [34] or "sewage load" [35]. As an example of a wrong shift in meaning, we can point to a definition used in [34] "Grey water footprint refers to the amount of water needed to absorb sewage generated by human

activity". However, such a shift in the meaning of the definition can lead to incorrect equations for calculating the grey water footprint, as was already documented in several articles [25].

The 3.5% proportion of serious errors in citations in the period from 2011 to 2023 can be considered as low. However, as can be seen from Table 1, six out of seven serious errors appeared in recent years. While there are no serious errors in the period before 2020 (with one exception), they occur every year between 2020 and 2022, and the error rate in the period 2020–2023 is 6.0%. It should be emphasized that due to the date of data collection, the data for 2022 and 2023 are preliminary. It can be expected that a generation of researchers emerges who do not study the grey water footprint as such but apply the grey water footprint as an established method in their own research. The probability of the wrong citation among these researches is higher, which is also evidenced by three cases where a book review was cited instead of citing the source book. It can be considered as a possible proof of the status described by Simkin and Roychowdhury [36,37], i.e., that the authors of the articles did not read some of the cited works, but copied from the lists of references used in other papers.

The emergence of a new generation of researchers using the grey water footprint can also be inferred from the proportion of references that were not written by the originators of the principle and definition of the grey water footprint. These references do not give credit to the originators of the idea, but, usually, just specify the source from which the authors of the article drew information. Although this share is less than 10% of all references, these references appear only in recent years. As shown in Figure 2, these references do not occur in the first period (up to 2017). Between 2017 and 2021, they occur in single cases (max three), and only in 2022 (although these are preliminary data) do they have a higher representation (11 citations from 39). Therefore, it can be expected that the number of these references will increase similarly to the case of citation errors.

One more interesting fact emerged from the analysis of citation errors. Four out of six miscited references were written by the originators of the grey water footprint definition or the grey water footprint principle. The citing authors most likely tried to give some credit to the authors of the grey water footprint principle and definition but chose the wrong article to cite.

The ratio between cited and uncited definitions of the grey water footprint is also interesting from the point of view of who is citing it. In articles written by the originators of the grey water footprint definition or principle, the source article was cited 16 times, and 14 times the definition was given without citing the source article. Other authors cited the definition of the grey water footprint 188 times (though 20 times they did not cite the originator of the definition or principle and once the cited reference was missing) and did not cite the source of the definition 72 times.

## 5. Conclusions

The study shows that incorrectly referenced sources represent only 3.5% of cited sources of the certain specific idea or method, in this case the definition of the grey water footprint. At the same time, the study shows that these errors increase with time passing since the concept was published. Similar, statistics apply to the number of citations of so-called "empty" references, i.e., citing someone else other than the originator of the idea. The proportion of these "empty" citations is less than 10% of the total number of citations of the grey water footprint definition, but also has been increasing in recent years. We explain this situation as the probable emergence of generation of researchers who do not study the grey water footprint idea as such, but use it as a tool for their own research mainly focused on other problems.

**Supplementary Materials:** The following supporting information can be downloaded at: https://www.mdpi.com/article/10.3390/publications11010008/s1, Table S1: Who is cited—References; References [19,21–24] are cited in the supplementary materials.

**Author Contributions:** Conceptualization, L.A.; methodology, L.A.; validation, L.S.; data curation, L.A.; writing—original draft preparation, L.A.; writing—review and editing, L.A. and L.S.; funding acquisition, L.S. All authors have read and agreed to the published version of the manuscript.

**Funding:** This research was funded by T. G. Masaryk Water Research Institute (Výzkumný ústav vodohospodářský T. G. Masaryka) in Prague, Czech Republic; via the internal grant number 3600.52.01/2022.

**Institutional Review Board Statement:** Not applicable.

**Informed Consent Statement:** Not applicable.

**Data Availability Statement:** The data presented in this study are available upon request from the corresponding author.

**Acknowledgments:** We would like to thank the anonymous reviewers for their insightful comments and recommendations during the peer review process.

**Conflicts of Interest:** The authors declare no conflict of interest.

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
