# Peer review of "Citation Accuracy: A Case Study on Definition of the Grey Water Footprint"

_publications, doi:10.3390/publications11010008_

Round 1

Reviewer 1 Report

The paper analyses citations to and quotations of the definition of grey water footprint (a technical indicator of water pollution) in the body of 204 articles retrieved from the Scopus database in order to find the percentage of misquotations and citations attributed to wrong authors. The study innovates by approaching the problem of citation and quotation errors in scientific literature through tracing the quotation content over the years. The research design and the methods used are innovative, technically sound, and clearly stated. Although data and results are consistent, they do not represent the major contribution of the paper, for they lead to simple conclusions. Nevertheless, the study is relevant for the field for its methods and novelty.

Specific comments:

The two figures mentioned in the manuscript could not be found in the body text or as supplementary material. They do not seem relevant, though, since it was possible to understand the content of the paper without them.

I suggest the authors point out to readers that data regarding the year 2023 is not complete and therefore should be interpreted with caution.

In line number 152 it is not clear if the authors are referring to DOI/URL of citing papers or cited papers.

Reviewer 2 Report

Abstract.

1.       “According to some studies, the citation error rate reaches tens of percent.” I don’t think this sentence should be placed in the abstract without supporting literature indicated.

2.      “However, these are often insignificant formal discrepancies that do not prevent tracing the source used.” Can you clarify what this sentence means?

3.      “The grey water footprint is part of the water footprint concept introduced in 2002 but was added to the water footprint concept later on.” If the grey water footprint is already a part of a concept, how can that be added to that concept again? Please be more accurate about their relationship.

4.      “Although this is a low number, these significant errors have been appearing only in recent years. This suggests that the percentage of errors could gradually increase as the use of grey water footprint expands in practice.” From Table 1, I see major errors have appeared since 2014, which is not a recent year. I also don’t think there is enough evidence in this study suggesting that the errors will “gradually increase,” as the error numbers range only from 1 to 2.

5.      “the number of cited works that were not written by authors of the water footprint concept and the grey water footprint definition, will probably increase.” No evidence of this trend is provided here. I suggest stating your findings based on the data instead of making a prediction of the future.

Introduction

1.      L24-29. Ironically, there is no reference here, even though the purpose of this paragraph is to emphasize references.

2.      L45-47. I don’t think it is necessary to say the “seed.” Instead, it will be better to add statements about how popular and important the concept of GWF is for a specific discipline and how dire the consequences will be if we ignore the citation errors for this concept. These statements will strengthen your study’s contribution.

3.      I see 1.1, but I don’t see the subsequent orders.

4.      L75,78. Abbreviated phrases should be written in full the first time that they are used, with the abbreviation in brackets. Don’t interchangeably use full terms and abbreviations.

5.      L81. The authors give a list of “correct quotation of originators of grey water footprint.” I don’t suspect the accuracy of the list. However, how should we ensure that the list contains all classic literature about GWF without omission? In other words, the precision rate may be high, but the recall rate is unknown. A way to solve this is to plot a citation tree/citation map to track the GWF literature history and highlight the “roots.”

6.      L102. What do you mean by misquotation? It sounds like the mistake of incorrectly repeating what the cited sources say, but from the context, I think it is intended to say citing the wrong source.

7.      L103. The authors need to explain the reason for asking the second question. It is not academic misconduct, and it is possible that the subsequent studies cite an improved or contextualized version of GWF different from any of the listed “must-cites.” I agree that this question has something, as Robert K. Merton talked about the idea of “obliteration by incorporation” in Social Theory and Social Structure, meaning that the originators and the literature are forgotten and rarely cited due to prolonged and widespread use. However, the authors should base the question on this context and explain more about its potential significance.

Materials and Methods

1.      L114. I don’t see figures in this manuscript, which makes the evaluation of figures impossible.

2.      L126- 127. Please give full terms of DOI and EID and explain their meaning to the audience who don’t know them.

3.      L146. I am not sure why articles that mention the definition of the grey water footprint but do not cite the source of the definition are excluded. This is a serious citation error if the article authors do not raise the definition of GWF. Also, 86 is not a small number; including them in the analysis may give rise to interesting results.

4.      L153. No need to mention Google Table.

Results & discussion

1.      Table 1. There should be data and discussion on the papers citing second-hand literature rather than the original papers defined in 1.1 because this part is paper-level while the following part is author-level.

2.      An analysis of how papers having citing issues are further cited by subsequent papers in the definition part will be interesting and helpful to see its impact. Again, plotting a citation tree can help.

3.      Table 2. Please use periods as decimal separators to avoid confusion.

4.      I can’t find Figure 2.

5.      L211~. Because this study already adopts human screening as the major method, “maximum objectivity” is not a good reason here to exclude all 86 articles. Inspectors should be able to judge whether a definition is given a citing source by its context, and this is the advantage of using human screening. At least, the authors should differentiate the “seemingly not citing” papers from the “totally not citing” papers and report their numbers.

6.      L221~. I see in the method that “the source documents were analysed to see if they contain the definition of the grey water footprint in accordance with the Water Footprint Assessment Manual, with the guidelines for Tier 1 or the principle of the grey water footprint described in the article by Chapagain et al. [22].” Does not this involve “assessing the accuracy of the adopted definition”?

7.      L245. To make this prediction, the current study is not enough. The trend is not increasing, and it is possible that the small number of incorrect citations are just outliers.

Reviewer 3 Report

In “Citation Accuracy: A Case Study on Definition of the Grey Water Footprint” the authors analyzed whether paper references in the Gray Water Footprint field correctly define the concept in question. I think that the data obtained need to be more explored so that conceptual errors are made explicit. Also, the reason why the research was carried out and its contribution to the area need to be clarified.

Specifically, I have the following suggestions:

1.     L100: It was not clear why the research was carried out. The authors should develop more on this point. For example, were the flaws in the conceptual definition relevant enough to generate any concrete problem in understanding the theme?

2.     L141: “In the next screening step, 56 articles that did not contain a verbal definition of water footprint were excluded”. I suggest that the authors explain how this exclusion was carried out, showing examples. Could this step contain some kind of bias?

3.     L153: “If more than one reference document was cited, then the oldest reference in which the grey water footprint definition was found was used”. I suggest clarifying this criterion used, because in some cases, the oldest reference may not yet contain the definition established by the field.

4.     L175: “Serious errors thus represent 3.5% of citations.” Perhaps, to understand the representativeness of this number in relation to this area, the authors could compare it with this same type of error already found in other areas. Indeed, errors in reference articles indexed in Scopus are not uncommon.

5.     L187: “A total of 90.15% of references are so-called "empty" references that do not contain an original idea under investigation, but strictly refer to other studies to substantiate their claim.” I really didn't understand what that means. Wouldn't it be the opposite?

6.     L166: The 42 references obtained in the last step shown in the methodology deserve a deeper analysis in the results.

7.     L229: “However, as can be seen from Table 1, six out of seven serious errors appeared in recent years and the error rate in the period 2020-2023 is 6.0%.” We are talking here only about the year 2022, whose analysis could not even be closed since the data were obtained in Oct/2022. In other years it oscillates at 1, 2 and 3, which is perfectly reasonable. I suggest that the data referring to 2022 be better explored.

Round 2

Reviewer 2 Report

The authors addressed the concerns well. I just have 3 points that need attention.

1.      The sentence “The citation errors range from several percent 8 to tens of percent” may be inappropriate in the abstract because it is not a result supported by this study, and I believe there is no consensus on it.

2.      The authors should recheck the use of “cite/citation” and “quote/quotation”.

3.      Table 3 is a little lengthy. Maybe it can be put in a supplement or something like it.

Author Response

Reviewer:

  1. The sentence “The citation errors range from several percent 8 to tens of percent” may be inappropriate in the abstract because it is not a result supported by this study, and I believe there is no consensus on it.

Response:

Thank you for your comment, we agree that this is not the result of our research and therefore specific values may not be given in the abstract.

Reviewer:

  1. The authors should recheck the use of “cite/citation” and “quote/quotation”.

Response:

Thank you for your comment. We have again checked the text of the article and corrected cases where the meaning of both terms as defined in the introductory paragraph was confused, i.e. citation errors as errors in referring to cited sources and quotation errors as errors in misquoting the content of cited sources.

Reviewer:

  1. Table 3 is a little lengthy. Maybe it can be put in a supplement or something like it.

Response:

We agree that the table makes the article less clear and has therefore been moved to supplementary material.

Finally, we would like to thank Reviewer 2 for a number of insightful recommendations that improved our paper.